# Non-Genetic-Induced Zebrafish Model for Type 2 Diabetes with Emphasis on Tools in Model Validation

**DOI:** 10.3390/ijms25010240

**Published:** 2023-12-23

**Authors:** Olakunle Sanni, Thandi Fasemore, Pilani Nkomozepi

**Affiliations:** Department of Human Anatomy and Physiology, University of Johannesburg, Doornfontein 2028, South Africa; tfasemore@uj.ac.za (T.F.); pilanin@uj.ac.za (P.N.)

**Keywords:** type 2 diabetes, animal model, zebrafish, hyperglycemia, insulin resistance

## Abstract

The unrelenting increase in the incidence of type 2 diabetes (T2D) necessitates the urgent need for effective animal models to mimic its pathophysiology. Zebrafish possess human-like metabolic traits and share significant genetic similarities, making them valuable candidates for studying metabolic disorders, including T2D. This review emphasizes the critical role of animal models in diabetes research, especially focusing on zebrafish as an alternative model organism. Different approaches to a non-genetic model of T2D in zebrafish, such as the glucose solution, diet-induced, chemical-induced, and combined diet-induced and glucose solution methods, with an emphasis on model validation using indicators of T2D, were highlighted. However, a significant drawback lies in the validation of these models. Some of these models have not extensively demonstrated persistent hyperglycemia or response to insulin resistance and glucose tolerance tests, depicted the morphology of the pancreatic β-cell, or showed their response to antidiabetic drugs. These tools are crucial in T2D pathology. Future research on non-genetic models of T2D in zebrafish must extensively focus on validating the metabolic deficits existing in the model with the same metabolic defects in humans and improve on the existing models for a better understanding of the molecular mechanisms underlying T2D and exploring potential therapeutic interventions.

## 1. Introduction

Diabetes is a chronic disease of metabolism that affects more than 537 million people in the world, and the number is projected to increase to 537 million by 2045 [1]. More than 90% of diabetes cases were diagnosed as Type 2 diabetes (T2D) [2]. T2D is characterized by a persistent increase in blood sugar (hyperglycemia) as a result of the impaired action of insulin at the target tissue coupled with insufficient insulin secretion by the pancreatic β-cells. Persistent hyperglycemia over the years could activate physiological pathways that often result in both micro and macrovascular complications. These complications accounted for 6.7 million people’s deaths, and the number is projected to increase to 73.6 million in 2045 [3]. At present, there is no definite cure for diabetes, but it can only be managed, hence the importance of diabetes research and the need to find a perfect animal model that reflects the pathophysiology of T2D in human.

Over the years, animal models have provided new insights into the mechanism of β-cells dysfunction, insulin resistance as well as diabetic complications and have provided the basic information required in the clinical translation of new drug targets. For example, metformin, the most widely prescribed antidiabetic drug in the world [4], was discovered for its sugar-lowering action in rabbits in 1929 and the first trial of metformin on humans was for the treatment of diabetes in 1959 [5,6]. However, the percentage of the new drug candidates who succeeded during clinical trials was very low due to the poor initial screening and assessment on inequivalent animal models. The most commonly used model is the rodent-based model [7]. However, the high cost, labor-intensive nature, and ethical issues presented a major setback for the continuous use of the rodent-based model. Nevertheless, the rodent-based model has provided a wide ranging advance tool in T2D research such as genetic modification and metabolic phenotype assessment [8,9].

The zebrafish model has continued to attract interest in diabetic research as an alternative to the rodent-based model. Recent research has reported the structural and physiological similarities in energy metabolism between zebrafish and humans [10]. Also, zebrafish has been found to offer a system for the discovery and characterization of new diagnostic and therapeutic targets for metabolic disorders such as obesity [11], non-alcoholic fatty diseases [12], atherosclerosis [13], and diabetes [14,15]. 

An animal disease model should be able to replicate a similar pathophysiology of the disease condition in human. For example, an animal model of T2D should be able to validate the critical biochemical mechanism of insulin resistance alongside with partial pancreatic β-cells dysfunction [9]. A wide array of literature has described the successful establishment of zebrafish models of T2D using different methods with each having its own advantages and disadvantages. As a result, this review looks at the methods of establishment of different type of zebrafish models of T2D with the emphasis on the model validation using indicators of T2D.

## 2. Material and Methods

Scopus, PubMed, and Google Scholar were used as search engines to retrieve published data. The keywords used for the search include “zebrafish model of type 2 diabetes”. A total of 101 articles were obtained (published between 2007 and 22 September 2023), out of which only ere selected based on the inclusion and exclusion criteria below. The result is summarized in Table 1 below. 

We used the following as inclusion criteria for the selection. 

Research articles that report the non-genetic model of type 2 diabetes in zebrafish.Research articles that report the hyperglycemic model in a zebrafish model.Research articles that report the obesity model in zebrafish.

The exclusion criteria were set to exclude the following articles.

4.Articles that report other animal models apart from zebrafish.5.Articles that report the zebrafish model of diabetic microvascular complications of T2D.6.Articles that reported the genetic model of T2D in zebrafish.

## 3. Advancement of Animal Models in Therapeutical Approaches of T2D

The animal model plays an important role in the success of any therapeutic agent in clinical trials and the use of an inequivalent animal model in preclinical assessment determines the percentage success achieved by the drug during clinical trials [27]. Therefore, it is imperative that the chosen animal model for drug discovery is able to answer the specific question on the pathophysiology of the disease and shows equivalency with human disease to fulfill the purpose of preclinical stages as well as bridge the gap of preclinical research and clinical trial. Few animal models for the study of therapeutical agent of T2D have translated to clinical studies. Some of these clinical studies are discussed as follows:

*Resveratrol*: Resveratrol (3,5,4′-trihydroxystilbene) is a bioactive compound present in different plant species. Evidence from animal studies demonstrated the therapeutical action of resveratrol in T2D using different mechanisms such as improvement of insulin action in animals with experimentally induced insulin resistance [28,29,30] and enhancing the insulin signaling pathway in the liver of insulin-resistant animals through the phosphorylation of several signaling proteins, including IRS-1, Akt and PI3K [31,32]. Preliminary clinical trials also show that resveratrol decreases insulin resistance in T2D patients, and they appear to be linked to changes in SIRT1 and AMPK [33,34,35,36], which therefore improves glycemic control. 

*Salicylates:* Salicylates is an anti-inflammatory drug but repurpose as new treatments for diabetes. Preclinical studies of salicylates in animal models were performed with Zucker fatty rats and ob/ob mice. The high dose of salicylates in these obese and severely insulin-resistant mice models significantly lowered blood glucose concentrations, improved glucose tolerance, and increased insulin sensitivity [37]. Pilot clinical trials to test the efficacy of salicylate were conducted in small groups of patients for 4 weeks. The results were similar to the ones obtained from the animal model. The high dose of salicylate increases insulin secretion [38], reduces metabolic clearance of insulin, and consequently increases circulating insulin concentrations. The repurposing of salicylates is currently funded by the NIH for larger clinical trials for both type 2 diabetes and cardiovascular disease.

*Lixisenatide*: Lixisenatide is an incretin hormone GLP-1 secreted by intestinal endocrine in response to food and enhances meal-stimulated insulin secretion, which is called the incretin effect [39]. The development of lixisenatide as a once-daily therapy is currently ongoing in an extensive Phase III clinical study in T2D patient [40]. The preclinical studies in animal models highlighted that lixisenatide enhances glucose-stimulated insulin secretion (GSIS) in a strictly glucose-dependent manner [41]; therefore, it has the potential to avoid undesirable hypoglycemia.

### Zebrafish as Disease Model

Zebrafish have increasingly gained attention as a model organism for biomedical research such as organelle biology research [42,43], developmental disorders research [44,45], mental disorders research [46,47], and communication between the brain and organs research [48]. Zebrafish have been shown [49] to possess the potential of a model organism for drug discovery, exploring mechanisms of drug action in vivo, and preclinical drug repurposing: hence its emergence as a leading model organism in human disease. Also, zebrafish have human-like metabolic traits that can be used to supplement information. Recent investigations [12,50,51] have clearly demonstrated this potential in which approved drugs that alleviate metabolic syndromes in humans are equally effective in the zebrafish model.

In addition, the genetic information of zebrafish is well documented as having the advanced capacity to accelerate the genetic investigations by gene overexpression or knockdown [52]. A direct comparison of the genomic structure of zebrafish and humans revealed that approximately 71.4% of human genes have at least one zebrafish orthologue and 69% of zebrafish genes have at least one human orthologue [53]. Interestingly, the tissue development and homeostasis of the muscle, liver, pancreas, spleen, blood, kidney intestine, bone and neurons in zebrafish is the same as that in humans [49].

Furthermore, the embryonic and adult stages of zebrafish can be applied to model human disease. The zebrafish embryos are transparent and fertilized ex vivo, providing efficient research in developmental toxicity [49]. They are available and convenient for protein target screening, transgenic, and gene editing [54]. The transparency of the embryos supported the real-time observation of the phenotype of drug activity on tissues or organs at single-cell resolution such as fluorescent in situ hybridization, immunohistochemistry, or tissue-specific fluorescent transgene expression [49]. On this note, the zebrafish larvae has been widely used in cardiovascular research as a result of the clear visibility of the heart coupled with active blood circulation enabling the accurate quantification of heart rate and rhythm in non-anaesthetized conditions [55,56]. 

## 4. Zebrafish as Metabolic Disease Model

Acquiring balance in energy intake, utilization and storage is a critical component of metabolic control and energy homeostasis that involves complex interactions between endocrine signaling and multiple organs. Therefore, a good metabolic disease model should be able to replicate the complexity of the metabolic and energy homeostasis in a multicellular context. The major organs related to metabolic regulation are evolutionary conserved and similar to that of humans. For example, the zebrafish pancreas comprises both exocrine and endocrine parts similar to those of humans, and they are able to produce insulin and glucagon [57,58]. Also, the hypothalamus that controls the appetite circuits and the insulin-sensitive tissues such as liver, muscle and adipose tissue are all present in zebrafish. Previous study on the comparative genomics of carbohydrate/glucose metabolic genes between fish and mammals revealed that 57.31% of human carbohydrate metabolic genes are involved in T2D as compared with 58.66% in zebrafish [59]. The genes are involved in 15 carbohydrate metabolic pathways and six key regulatory pathways, including carbohydrate digestion and absorption, insulin secretion, insulin resistance, the insulin signaling pathway, the glucagon signaling pathway, and the adipocytokine signaling pathway [60,61]. These similarities in carbohydrate/glucose metabolism offer insightful models for not only researching the fundamental biology of glucose metabolism but also exploring the mechanisms of human metabolism.

Differences exist between zebrafish and humans that limit zebrafish as a model for metabolic diseases. For example, metabolic diseases involve lifestyle, socioeconomic, and behavioral factors that are unique to humans. Therefore, dietary interventions are difficult to translate to human dietary recommendations due to fundamental differences in macro- and micronutrient requirements [62]. Also, in the zebrafish model, it is difficult to track individual food intake. Food intake is closely linked to growth rate, which in turn influences adipocyte formation [63]. Other differences include temperature, which affects metabolic rate and body fat composition, as zebrafish are poikilothermic animals [64]. 

Nevertheless, zebrafish has emerged as a perfect model for the screening of metabolic diseases due to its accessibility to embryonic and genetic modifications, optical transparency of its embryos, relatively quick generation time, and simple maintenance. As a result of this, researchers have developed zebrafish models (summarized in Table 1) that provide powerful tools for understanding pathophysiology and possible intervention for the common metabolic diseases. This review briefly discussed the following metabolic disease model in zebrafish due to similar disruptions in normal metabolism similar to T2D. 

### 4.1. Obesity

Obesity is as a result of imbalance energy metabolism that resulted in energy intake far greater than energy expenditure. Diet-induced and genetically modified approaches have been employed by researchers to develop zebrafish models of obesity. The diet induces obesity in zebrafish, and obese mammals have been shown to have similar pathophysiological pathways [12,65]: hence the wide use of the model to identify pharmacological targets and discover novel drugs for the treatment of obesity in humans. 

However, the transgenic obesity model in zebrafish is developed through the overexpression or knockout genes that cause obesity in mammalian models. For example, the overexpressing AgRP (*Tg(b-actin:AgRP*) was used to develop a transgenic zebrafish obesity model that demonstrated the phenotype of obesity in mammals such as body weight increase, linear growth, visceral adipose accumulation, and increase total triglycerides [66,67]. The transgenic zebrafish model was also developed by the gene knockout of miR-27b, and the model demonstrated hyperlipidemia, hepatic steatosis and increased white adipose tissue mass similar to a human model.

### 4.2. Non-Alcoholic Fatty Liver Disease

Non-alcoholic fatty liver disease is a metabolic disease characterized by the accumulation of fat droplets in hepatocytes as a result of several factors such as lipid dysregulation, mitochondrial dysfunction, inflammation, de novo lipogenesis and oxidative stress [68,69]. The tissue structure and function in zebrafish is similar to those of mammals as well as the pathophysiology of steatosis that gradually develops into NASH [12]. Hence, researchers have developed the zebrafish model to understand the disease in humans.

Currently, three approaches have been employed to develop zebrafish models of NAFLD: these are dietary, chemical, and genetic modification. The dietary approach of NAFLD models includes zebrafish fed with a high fat diet, high-cholesterol diet, and fructose diet (Table 2). The chemical approach involved the treatment of the zebrafish with hepatotoxic agents. The hepatotoxic agents that have been reportedly used are tunicamycin [69], thioacetamide [70] and valinomycin [70].

### 4.3. Cardiovascular Disease

Growing evidence has highlighted the importance of zebrafish as a vertebrate model for cardiovascular disease research. The vascular anatomy and heart rate of zebrafish is highly conserved and more similar to human than those of other model organisms [81,82]. Also, the optical transparency of the embryos made it possible to observed non-invasively the morphological changes of the heart and the functional changes of blood vessels in living animals [83]. The genetic modification approach is more precise and effective in developing the cardiovascular disease model in zebrafish. In contrast, the diet-induced approach only mimics the deleterious effect of high lipids on the cardiovascular system such as fatty streak formation in the arteries, vascular lipid accumulation, and the endothelial cell layer disorganization and thickening.

## 5. Type 2 Diabetes Model of Zebrafish

Several non-genetic-induced zebrafish models have been developed over the years with the aim of replicating the disease pathophysiology in humans using different procedures. These procedures can be classified as follows. 

### 5.1. Glucose Solution Method

This method of inducing hyperglycemia is widely used in the zebrafish model. The adult or larvae are immersed in glucose solution for a specific period of time. Different solutions were used by different researchers to induce hyperglycemia. Some researchers use a gradational glucose solution at different times, as summarized in Table 3.

### 5.2. Diet Induction Methods

Obesity is the leading risk factor of T2D, as over 90% patients with of T2D are obese or overweight [84]. Therefore, inducing obesity would show the classic symptoms of T2D such as hyperglycemia, hyperinsulinemia and eventually insulin resistance. The mechanism involve the activation of NF-kβ signaling that promotes insulin resistance and pancreatic β-cell dysfunction [85,86]. Various experimental procedures have been developed to induce T2D in zebrafish using the diet-induced method, which are summarized in Table 4.

### 5.3. Chemical Induction Methods

Alloxan and streptozotocin have been widely used in rodent models of T2D. However, the use of these chemicals in the zebrafish model is only documented in type 1 diabetes [15,23,87]. Bisphenol compounds have been reported to modulate glucose metabolism [24]. Furthermore, recently, two studies have demonstrated the use of bisphenol compounds to induce a zebrafish model of T2D. The studies exposed different derivates of the bisphenol compound to zebrafish larvae for 28 days [24,25]. At 10 µg/L, each derivate demonstrated similar diabetogenic effects such as insulin resistance, decreased level of plasma insulin, and impaired glucose homeostasis, while hepatic gluconeogenesis and glycogenolysis were promoted.

### 5.4. Combined Diet-Induced and Glucose Concentration Methods

T2D is a complex and multidimensional metabolic disease, and most of the models discussed above present one flaw or the other; therefore, approaches were made by researchers to find alternative models that will be consistent with the T2D conditions in humans. Hence, they combined diet-induced and glucose concentration methods.

Wang et al. [26] were the first to develop a combined diet-induced and glucose concentration method. They exposed zebrafish larvae to 2% glucose concentration and at the same time fed the larvae a high-cholesterol diet (10% cholesterol). This model produced a manifestation of the diabetes model in rodents. Also, it accounted for most of the flaws in each of the models (i.e., glucose concentration and diet-induced methods). For example, persistent hyperglycemia was lacking in the glucose concentration method but was possible in this combined method. Likewise, the longer induction time in the diet-induced approach (usually 6–10 weeks) was greatly reduced to (2 weeks). The same method was applied to adult zebrafish with little modification by the same author using 3% glucose concentration and the same content of cholesterol in the diet for 19 days [88]. The result of this model resembled the phenotype of T2D in humans.

## 6. Tools Used in Zebrafish Type 2 Diabetes Model Verification

T2D is characterized by hyperglycemia as a result of insulin resistance and pancreatic β-cell dysfunction [89]. Therefore, any animal model developed to mimic T2D in humans should be able to demonstrate the failure of cells to respond to insulin, β-cells not producing enough insulin, and persistent high blood glucose. As a result, tools such as persistent hyperglycemia, the insulin resistance test, the glucose tolerance test, pancreatic β-cell morphological examination and the antidiabetic drug response test are often employed to validate animal models of T2D (Table 5). 

### 6.1. Persistent Hyperglycemia

Persistent hyperglycemia is the hallmark of T2D [19]. A good model of T2D must be able to demonstrate high blood glucose over a period of time. The glucose concentration methods described in Section 4.1 demonstrated a persistent hyperglycemia after the withdrawal of glucose solution. For example, the model by Capiotti et al. [18] increased the blood glucose level up to 4–5 fold in comparison to the control group, but the glucose level decreased by 68% after 7 days of glucose withdrawal. In contrast, the diet-induced method demonstrated persistent hyperglycemia from the 1st week of the diet-induced approach up to 8 weeks. However, after 4 weeks of calorie restriction, the hyperglycemia was still persistent [21]. The chemical-induced method also show persistence, but there was only a marginal increase compared with the control.

### 6.2. Insulin Resistance Test

Insulin resistance is a crucial feature of T2D. The estimation of insulin resistance is used to assess the clinical state of T2D and responses to interventions. The widely used insulin resistance tests in animal models are the Homeostasis Assessment Model (HOMA), insulin tolerance test (ITT), and glucose tolerance test. HOMA a valid mathematical model to assess peripheral insulin sensitivity and β-cell function using the value of fasting blood glucose and serum insulin concentration [90]. ITT is used to obtain the estimate of insulin resistance from the rate at which glucose declines after an intravenous injection of insulin. The test reflects the interaction between insulin stimulation of peripheral glucose absorption and suppression of the liver’s production of glucose [91]. Glucose tolerance could be measured as oral (OGTT) or intraperitoneal glucose (IPGTT) load 2 h afterwards to obtain insulin resistance from the logarithm of the plasma insulin concentration [91]. It is employed to assess insulin resistance and apparent insulin overload. 

Using the larvae stage of the zebrafish as a T2D model posed a challenge regarding using the insulin resistance test due to the volume of the blood, since it is difficult to measure blood glucose by the collection of blood at different time intervals [92]. Consequently, the insulin resistance test was not reported in the model of zebrafish where larvae are used. In the adult zebrafish model, the insulin resistance test was not reported in any of the glucose concentration methods. This is probably due to the fact that the hyperglycemic state is transient and constant immersion in glucose solution, which is required to maintain the hyperglycemic state, will not provide a basal state for the measurement in OGTT or IPGTT.

However, the insulin resistance test was reported in the diet-induced model. For example, Zang et al. [21] performed an ITT as well as both IPGTT and OGTT, and the result showed similar resemblance to diabetic data in humans and rodent models of T2D. Unfortunately, the insulin resistance test was not reported for the combined diet-induced method with glucose concentration and the chemical-induced method. 

### 6.3. β-Cell Morphological Examination

T2D is characterized by hyperglycemia due to insulin resistance followed by β-cell loss as a result of β-cell exhaustion and glucotoxicity, resulting in β-cell dysfunction. Hence, β-cell morphology formed an important tool for T2D model verification. Unfortunately, a β-cell morphological examination was not reported in any of the approaches for the zebrafish T2D model. However, β-cell mass was only measured in the novel zebrafish model for T2D using diet-induced obesity as an indicator of compensatory β-cell hypertrophy and hyperplasia in response to hyperglycemia in diabetogenic states in zebrafish similar to humans [93]. 

### 6.4. Antidiabetic Drug Response Test

A positive response to human antidiabetic drugs is an important step in validating any animal model of diabetes. It also validates the metabolic deficits existing in the model with the same metabolic defect in humans and predicts the response to the screening of new antidiabetic drugs. 

Sulfonylureas, biguanides and thiazolidinediones are classes of antidiabetic drugs whose mechanisms of action, such as enhancing insulin action and secretion, describe the pathophysiology of T2D [94]. These classes of drugs have been used to validate the T2D model of zebrafish. For example, the glucose solution method by Mohammadi et al. [95] used metformin, a biguanide, for their antidiabetic response test. Zang et al. [21] in their diet-induced method used glimepiride, a sulfonylureas, and metformin to validate their model. Pioglitazone was also employed in an antidiabetic drug response test in a combined diet-induced and glucose solution method [26]. However, the antidiabetic drug response test in the chemical-induced method is not clear enough, as the antidiabetic drug rosiglitazone was incubated along bisphenol compounds [24], which can provide a protective effect that is not curative.

## 7. Conclusions and Future Perspectives

There are challenges that come with using zebrafish as a model for T2D, such as unstable and inconsistent hyperglycemia, the sometimes toxic impact of chemicals and doses used for induction, and the inability to encompass different disease stages considering the remaining functional β-cell mass, prompting a continuous quest for an ideal model that accurately mirrors T2D’s pathophysiology in humans. Despite the limitations, zebrafish have optical transparency during the initial developmental stages that can enable non-invasive imaging and the real-time monitoring of metabolic processes, allowing for the study of beta-cell functionality, glucose uptake, and the investigation of mechanistic and therapeutic aspects of antidiabetic medications. These advantages can be exploited for the T2D model but requires optimization. As highlighted in this review, there was an insufficient focus on validating the zebrafish T2D models across the five different approaches. To advance our understanding of the molecular mechanisms underlying T2D, future research must enhance the existing non-genetic-induced model of T2D in zebrafish by validating the metabolic deficits existing in the model with the same metabolic defect in humans in order to fill the gap between preclinical research and clinical trials.

## Figures and Tables

**Table 1 ijms-25-00240-t001:** The results of literature research.

S/N	Research Articles	References
1	Induction of hyperglycemia in zebrafish (Danio rerio) leads to morphological changes in the retina	[16]
2	Alternate immersion in an external glucose solution differentially affects blood sugar values in older versus younger zebrafish adults	[17]
3	Biochemistry PPB, Biology M: Persistent impaired glucose metabolism in a zebrafish hyperglycemia model.	[18]
4	Chronic hyperglycemia affects bone metabolism in adult zebrafish scale model	[19]
5	High glucose levels affect retinal patterning during zebrafish embryogenesis	[20]
6	Development of a novel zebrafish model for type 2 diabetes mellitus	[21]
7	Intracellular insulin and impaired autophagy in a zebrafish model and a cell model of type 2 diabetes	[22]
8	Effects of streptozotocin on pancreatic islet β-cell apoptosis and glucose metabolism in zebrafish larvae	[23]
9	Bisphenol S exposure impairs glucose homeostasis in male zebrafish (Dan-io rerio)	[24]
10	Acute toxicity, teratogenic, and estrogenic effects of bisphenol A and its alternative replacements bisphenol S, bisphenol F, and bisphenol AF in zebrafish embryo-larvae	[25]
11	Impact of a combined high cholesterol diet and high glucose environment on vasculature	[26]

**Table 2 ijms-25-00240-t002:** Summary of zebrafish model for metabolic diseases.

Disease Model	Mode and Method	Target That Replicates the Disease Complexity in Human	Key Findings	References
Obesity	Larvae fed with high-fat diet	Adiposity	The model increases in adiposity in the presence of obesogenic compound (tributylin chloride) and high-fat diet and a dose–response respond to anti-obesogenic compound	[51]
	Adult zebrafish fed with high-fat diet	Energy homeostasis and metabolism are articulated in organs, in response to appetite regulation, insulin regulation and lipid storage Regulatory factors of obesogenic pathways such as PPARα/γ and LEP are expressed in zebrafish	The model demonstrated body mass index increase as well as hypertriglyceridemia and hepatosteatosis The model revealed that SREBP1, PPARα/γ, NR3H1 and LEP are key regulatory factors in these pathways and are expressed in zebrafish and mammalian obesity	[11]
	Adult zebrafish fed with high-cholesterol diet	The mitochondrial fatty acid β-oxidation, 3-hydroxyacyl-coenzyme A dehydrogenase and citrate synthase activity in the liver and skeletal muscle are downregulated	The model had almost twice the body fat (visceral, subcutaneous, and total fat) volume and body fat volume ratio (body fat volume/body weight) than dose-fed low-fat diets	[71]
	Transgenic zebrafish	Ectopic fat deposition in other organs and adipocytes. Inflammatory signaling pathways are upregulated, leading to insulin resistance	The model demonstrated an overexpression of active *akt*	[66]
Non-alcoholic liver fatty diseases	Adult zebrafish fed with high-fat diet combined with egg yolk powder	The liver physiological processes such as endoplasmic reticulum stress and glucolipid metabolism are dysregulated	The model demonstrated increased liver lipid deposition along with massive steatosis and ballooning. Also, the liver cells showed massive apoptosis	[72]
	Larvae fed with high-cholesterol diet	Increased expression of inflammatory cytokines	The model increases the lipids in the whole body, especially in the abdomen, where the liver and adipose tissue are located	[73]
	Larvae fed with fructose	Activation of the target of rapamycin complex 1 (Torc1) pathway	The model increases hepatic lipid accumulation, mitochondrial functions abnormalities, and endoplasmic reticulum defects	[74]
	Larvae fed with fructose 1 µg/mL tunicamycin	Activation of hepatic stellate cells and impaired hepatocyte secretion	The model increases hepatic lipid accumulation and laminin deposition	[69]
	Adult zebrafish injected intraperitoneal with thioacetamide	Activation of oxidative stress and decrease SIRT1 mRNA expression	The model increase hepatic lipid peroxidation and the accumulation of lipids in the liver	[70]
	*yy1* transgenic larvae	De novo FFA synthesis	The model increases hepatic triglycerides accumulation and causes steatosis	[75]
Alcoholic liver fatty diseases	Adult zebrafish fed with 1% (*v*/*v*) ethanol solution	Upregulation of lipogenesis, liver steatosis, steatohepatitis, fibrosis, neoplasia, and tumor-related genes	The model demonstrated the early-stage characteristics of alcoholic liver steatosis and steatohepatitis such as fat droplet accumulation, ballooning, hepatocyte destruction, and Mallory body development	[76]
Cardiovascular	Transgenic zebrafish	Reduced contractility and aberrant cardiomyocyte shape, dilated and raptured blood vessels	The model demonstrated the rhythm of the heart causing, for example, a slow rate, a fibrillating pattern or an apparent block to conduction	[77,78,79,80,81]
	Adult zebrafish fed with high-cholesterol diet	Increase lipoprotein oxidation, and fatty streak formation in the arteries	The model demonstrated vascular lipid accumulation and the endothelial cell layer disorganization and thickening	[13]

**Table 3 ijms-25-00240-t003:** Glucose solution method for zebrafish T2D model.

Glucose Solution Concentration	Duration	Stage of Zebrafish	Remark	Reference
Immersion in 2%	30 days	Adult zebrafish (1–3 years)	Continual immersion in a 2% glucose solution induces transient hyperglycemia (434 ± 1.4 mg/dL) and fluctuates depending on glucose levels in environment	[16]
Gradational increasing of glucose concentration (from 1% → 2% → 3%)	2 months	Adult zebrafish (1–3 years) and young (4–11 months)	Gradational glucose concentration resulted in hyperglycemia that is persistent for 2 months in younger zebrafish adults	[17]
Immersion in 111 mM	14 days	Adult zebrafish	The model resulted in a 4–5-fold increase in blood glucose levels. However, the blood glucose level decreased after 7 days of glucose withdrawal, but it was still almost 2 times higher than the control group	[18]
Immersion in 4 %	28 days	Adult zebrafish	Induced transient hyperglycemia up to values greater than 500 mg/dL	[19]
	5 days	Larvae	Induced transient hyperglycemia up to values greater than 500 mg/dL	[20]

**Table 4 ijms-25-00240-t004:** Diet-induced methods of zebrafish T2D model.

Diet Formulation	Duration	Stage of Zebrafish	Remark	Reference
6-fold overfeeding per day with Otohime B2 containing 11% crude fat, 51% crude protein, 2.3% crude calcium, 1.5% phosphorous, a maximum of 15% ash, 3% crude fiber	8 weeks	Adult male zebrafish (4–6 months old)	After 8 weeks, the blood glucose level significantly increased from 46 ± 5 mg/dL to 68 ±11 mg/dL. The calories were restricted for 2 weeks, and the blood glucose level decreased (from 64 ± 11 to 48 ± 12 mg/dL	[21]
Fed twice daily with high-fat diet (1% egg yolk representing high fat diet) together with brine shrimp (60 mg cysts)	10 weeks	Adult zebrafish	Fasting blood glucose level was significantly higher and insulin resistance was induced.	[22]

**Table 5 ijms-25-00240-t005:** Comparison of tools used in different approaches for a zebrafish T2D model.

	Glucose Solution Method	Diet-Induced Method	Chemical-Induced Method	Combined Diet-Induced and Glucose Solution Method
Persistent hyperglycemia	No, the glucose level decrease significantly after glucose withdrawal	Yes, even after calorie restriction	Yes but marginal increase compared to the control	Yes
Insulin resistance test	No	Yes	Not reported	Not reported
β-cell morphological examination	Not reported	Not reported	Not reported	Not reported
Antidiabetic drug response test	Yes	Yes	Yes	Yes

## Data Availability

The data presented in this study are available in the article.

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
