# Peer review of "Non-Genetic-Induced Zebrafish Model for Type 2 Diabetes with Emphasis on Tools in Model Validation"

_ijms, 2023, doi:10.3390/ijms25010240_

Round 1

Reviewer 1 Report

Comments and Suggestions for Authors

The first line of the abstract is a non sequitur: it assumes that animal models will be lead to treatments for type two diabetes (T2D) but I don’t think they can state this without robust evidence to support it. I cannot find this evidence in the main text of the paper.

The authors focus on the genetic similarities between zebrafish and humans, which they believe make zebrafish valuable candidates for studying T2D. However, while there may be similarities it is important to acknowledge the differences, which are many. Even small differences may lead to significant differences in outcome, which is of course vitally important when developing drugs.

The review claims to be about T2D but the authors devote several pages of this short review to obesity, NAFLD, cardiovascular disease, with the T2D bit starting halfway through the paper. Perhaps the title of the review should be altered to more accurately reflect its contents? Or the authors should more clearly state why they are considering these aspects.

Their conclusion seems strange. They note that the use of zebrafish as a model for T2D model is problematic for various reasons - unstable and inconsistent hyperglycemia; the toxic effect and inconsistence dose of the chemicals for induction; inability to account for different stages of the disease through the remnant functional β-cell mass – factors which mean that the zebrafish cannot accurately replicate human T2D. Yet they bizarrely conclude that despite this, the zebrafish model is great because of the ‘optical transparency during early developmental stages which facilitates non-invasive imaging and real-time observation of metabolic processes’. In other words, they are acknowledging that zebrafish are not a good model for human T2D but because the fish are transparent it’s apparently ok, because we can study them easily and gain knowledge about zebrafish. This focus on internal validity at the expense of external validity is inappropriate for a field that aims to have relevance to human T2D. It seems that zebrafish are used due to their ‘accessibility to embryonic and genetic modifications, optical transparency of its embryos, relatively quick generation time, and simple maintenance’ rather than their ability to accurately replicate human diseases.

Comments on the Quality of English Language

English is generally good, some editing needed.

Reviewer 2 Report

Comments and Suggestions for Authors

Dear Authors,

This is a well-written and comprehensive review article on the Non-genetic model of T2D in Zebrafish. There is little literature available that comprehensively describes these models. The current work by Sunni et al will be useful for the field.

Upon careful review of the manuscript, I found the overall information very significant and well-presented. Though the review has cited a wide range of related articles some information is found missing in this review. Please see the comments below.

1.    Please shed some more light on the method. How was the literature survey performed? Any database searched? Providing information on the search query? (zebrafish OR danio) AND (diabetes OR diabetic OR hyperglycemia OR hyperglycemic OR glucose OR pancreas OR pancreatic OR insulin OR MODY) etc.

2.    As the authors described here, there are several similarities between the human and the zebrafish, Is glucose metabolism also similar between the zebrafish have similar?

3.    Please discuss the neuronal and hormonal systems development in these two systems.

4.    Though the non-genetic ways of diabetes induction may be more available, cheaper, or easier to induce, a genetic method may be more precise to target specific genes and produce abnormalities that are more accurate and more specific in results than other means. Please discuss in this review.

5.    Providing a comparison table to magnify the similarities and differences between the mentioned methods would be important to shed light on this method's significance.

6.    As the zebrafish model can be categorized as larval or adult stage. Please summarize whether both model holds similar characteristics.

7.    Achievements and disadvantages or challenges of using zebrafish as a model to study diabetic M should be discussed.

8.    As a part of the future direction, the authors should discuss measures that would facilitate finding therapeutic approaches for this syndrome for example, integration of the current modeling of T2D with new technologies, such as microfluidics by using zebrafish embryo, fish-on-a-chip, computational drug screening, and three-dimensional image analysis.

 I believe the comments provided above will add value to it and improve the manuscript significantly.

.

Round 2

Reviewer 1 Report

Comments and Suggestions for Authors

Comment 1

This comment is about whether animal models will lead to treatments for T2D. In response to my objection that we can’t assume that animal models will lead to treatments for T2D, the authors write ‘There is evidence in the manuscript that supports the use of animal models for therapeutical interventions. Kindly refer to paragraph 2 of the introduction.’

I have referred to para 2 as instructed, and find this: ‘Over the years, animal models, have provided new insights into the mechanism of β-cells dysfunction, insulin resistance as well as diabetic complications and have helped to elucidate new drug targets for the development of antidiabetic drugs.’ I am not disagreeing with this statement, but I am asking, as a result of these insights and elucidations, have animal models actually advanced the treatments for T2D? Have these insights resulted in benefits for T2D patients? If so, please outline how here. If not, this needs to be recognised. It’s not enough to continually assert than animal models are contributing knowledge and insights, if in fact these insights to do not translate into benefits for humans. By all means, point to the use of animal models for generating knowledge, but don’t claim that this knowledge leads to benefits for humans without any supporting evidence – that’s just an unfounded assertion.

Comment 2

This comment is about whether to emphasise the similarities or differences between humans and zebrafish. My point is that even small differences may lead to significant differences in outcome, which is of course vitally important when developing drugs. The authors reply that although animal models have a lot of differences compared to humans, they still play ‘a successful role in understanding the pathophysiology of the diseases and promoting therapeutic interventions in humans.’ (my emphasis). I agree that they may contribute to understanding the pathophysiology of the diseases in those animal models. However, I cannot agree with the second part of the sentence – that they promote therapeutic interventions in humans – unless the authors provide evidence in support of this assertion.  They continue, ‘In our opinion, the similarities between zebrafish and humans that make zebrafish suitable for the T2D model should be the focus rather than the differences. I hope you will support our opinion on this.’ I’m afraid I do not support this opinion. I believe that focusing on similarities at the expense of acknowledging important differences is contributing to the appalling failures of translation that we are seeing, and the huge unmet need in terms of treatments for patients.

Comment 3: I understand the authors’ response, but I feel they should spell out more clearly in the paper why they are considering these aspects.

Comment 4: The authors have not addressed my comment. They have simply added in something to the first sentence of the conclusion that results in a very long and confusing sentence. In terms of the limitations they simply conclude that ‘future research to mitigate those limitations was recommended.’ But there is no evidence that improving animal studies actually improves translation to humans. In fact there is evidence that it does not.

In general, I am not asking the authors to agree with me about this, I am simply saying that they need to tone down their rhetoric about clinical translation. If animal models lead to increased knowledge that’s fine, but don’t proceed to assert that they benefit humans without providing supporting evidence. It’s simply the difference between generating knowledge and generating clinical impact. Their review points to the former, so it will be much improved if they resist making claims about the latter.

Comments on the Quality of English Language

Generally good, needs some improvment

Reviewer 2 Report

Comments and Suggestions for Authors

Thank you for addressing all the comments.

I agree with the author's response in comments 3, 5 and 8.  The revised manuscript further improved the quality of this manuscript. I have no further comments.

Author Response

Thank you for taking the time to review our manuscript

Round 3

Reviewer 1 Report

Comments and Suggestions for Authors

I have twice recommended a major revision of this paper. Both times the paper has been returned within days with only minor revisions and with inadequate attention to the points raised. 

Comments on the Quality of English Language

Not as good as the first draft.
